# Effects of *Ganoderma lucidum Polysaccharides* on Dexameth-Asone-Induced Immune Injury in Goslings

**DOI:** 10.3390/ani15213226

**Published:** 2025-11-06

**Authors:** Huiying Wang, Guangquan Li, Xianze Wang, Shaoming Gong, Daqian He

**Affiliations:** Institute of Animal Husbandry and Veterinary Medicine, Shanghai Academy of Agricultural Science, Shanghai 201106, China; wanghy2010@189.cn (H.W.); lgqdx123@126.com (G.L.); wxz13187058389@163.com (X.W.); gsm117@163.com (S.G.)

**Keywords:** gosling, growth, *Ganoderma lucidum polysaccharides*, immune, serum biochemical markers, cecal microbial diversity

## Abstract

The study tested whether ganoderma lucidum polysaccharides (GLP) can reduce immune damage in goslings caused by dexamethasone (DEX), a substance that suppresses immunity. It used 180 one-day-old goslings split into three groups: a control group (no DEX, no GLP), a DEX group (given DEX, no GLP), and a DEX + GLP group (given DEX and 0.2% GLP in feed). From day 14 to 21, the DEX and DEX + GLP groups received DEX injections, while the control received saline. The team measured growth, immune organ health, blood parameters, organ structure, and gut bacteria at day 21 and 35. At day 21, the control group had better growth, stronger immune organs, and healthier blood markers than the other two groups. By day 35, the DEX + GLP group had better growth and improved immune/organ health compared to the DEX group. GLP also fixed DEX-caused damage to immune organs and balanced the gut bacteria (e.g., increasing helpful bacteria like Bacteroidetes). Overall, GLP effectively reduced DEX-induced immune injury in goslings and helped restore their growth.

## 1. Introduction

With the rapid development of intensive large-scale breeding in China, the frequent introduction and trade of livestock and poultry have led to the emergence of complex and recurrent diseases. Moreover, immunosuppression induced by various factors has imposed considerable challenges on the prevention and control of these diseases [1]. In this context, plant polysaccharides have garnered extensive attention due to their diverse beneficial properties, including antioxidant, antitoxic, anti-inflammatory, antibacterial, and immunomodulatory activities [2]. Among them, Chinese herbal polysaccharides have become a research focus because of their non-toxicity and potential pharmacological and biological effects. Investigating their mechanisms and impacts on organisms is of great significance for advancing livestock and poultry production toward high efficiency, health, and sustainable organic development [3]. Developing Chinese herbal preparations that combine immune-enhancing and growth-promoting effects has thus become a key strategy for disease prevention and control in the post-antibiotic era. As essential constituents of plant extracts, polysaccharides are widely found in animals, plants, bacteria, and fungi, exhibiting high biological activity in anti-tumor, antioxidant, immunomodulatory, and anti-inflammatory processes, and are recognized as crucial immunomodulators [4].

*Ganoderma lucidum polysaccharides* (GLP) are heteropolysaccharides composed of various monosaccharides, including glucose, galactose, and mannose, linked by glycosidic bonds, and possess complex and diverse structures such as β-glucan and α-glucan. This unique structural composition confers potent immunomodulatory properties, enabling GLP to enhance both non-specific and specific immune functions by stimulating macrophage activity, activating T and B lymphocytes, and promoting the proliferation and differentiation of immune cells [5]. In addition, GLP exhibit strong antioxidant effects, effectively scavenging excessive free radicals, reducing oxidative stress-induced cellular and tissue damage, and maintaining homeostatic stability in livestock and poultry. They also demonstrate notable anti-tumor and anti-inflammatory activities [6,7,8]. Importantly, the physiological effects of GLP are highly dependent on the intestinal microecological environment, as these active components are not easily digested or absorbed directly by the host, and their in vivo metabolism and efficacy largely rely on modulation by intestinal microorganisms.

A large number of microorganisms colonize the intestinal tract of animals, and the interaction between the intestine and its microbiota is a critical factor influencing growth, development, and overall health. These intestinal microorganisms play essential roles in nutrient digestion and absorption, the development of the immune system, maintenance of immune homeostasis, and resistance to pathogenic invasion [9,10]. Disruption of intestinal microbiota homeostasis can lead to host metabolic disorders and increase susceptibility to disease, with the diversity and richness of the microbiota being directly associated with animal health [11]. Although GLP have been investigated in basic research and applied explorations within certain livestock and poultry systems [12,13], studies focusing on gosling breeding, particularly addressing the issue of immune injury in poultry, remain limited. Therefore, this study aimed to provide a scientific foundation for the rational application of GLP in gosling production and to support the development of novel, green immunomodulators.

## 2. Materials and Methods

### 2.1. Animal Ethics Guidelines

This animal experiment was conducted following approval from the Animal Ethics Committee of Shanghai Academy of Agricultural Sciences (Approval No.: SAASPZ0522050), and all experimental procedures strictly adhered to the Guidelines for the Welfare of Laboratory Animals (GB/T 42011-2022) [14].

### 2.2. Experimental Feeding and Management

A total of 180 healthy one-day-old male goslings with normal mental status were randomly assigned to three groups, each with six replicates of 10 goslings, and no significant differences in initial body weight (BW) were observed among replicates (*p* > 0.05). The control (Con) and DEX groups were fed a basal diet, whereas the DEX + GLP group received a basal diet supplemented with 0.2% GLP. From days 14 to 21, goslings in the DEX and DEX + GLP groups were intraperitoneally injected with 3.5 mg/kg BW of DEX, while the Con group received 1 mL/kg BW of normal saline. The GLP used in the study was provided by the Institute of Therapeutic Standards, Shanghai Academy of Agricultural Sciences. This sample was sent to Jiangsu Edison Biotechnology Co., Ltd (Yancheng, China). for component analysis of GLP. The results showed that it mainly consists of galactose (Gal), fucose (Fuc), mannose (Man) and glucose (Glc) with a molar ratio of 0.500:0.121:0.133:0.246, among which galactose is the main constituent monosaccharide. Determined by gel permeation chromatography (GPC), the weight-average molecular weight (Mw) is 7243 Da, the number-average molecular weight (Mn) is 107Da, and the Z-average molecular weight (Mz) is 1.24 × 10^4^ Da. Experimental goslings were sourced from Anhui Xiangtiange Family Farm, and the breeding experiment was conducted at the Zhuanghang Experimental Station of the Shanghai Academy of Agricultural Sciences. The goose house was thoroughly cleaned and disinfected one week prior to the experiment, and throughout the study, goslings had ad libitum access to feed, water, and space for movement. Temperature, humidity, and ventilation were carefully controlled, and strict environmental hygiene was maintained. All feeding procedures, immunological interventions, and environmental management followed established standards [15]. The composition of the basal diet is detailed in Table 1.

### 2.3. Determination of Growth Performance

Goslings were weighed on an empty stomach at days 14, 21, and 35 to calculate the average daily gain (ADG). Daily feed intake was monitored by weighing the feed provided before 08:00 and the remaining feed at 20:00, allowing calculation of the average daily feed intake (ADFI). The feed-to-gain ratio (F/G) was subsequently determined based on the recorded ADG and ADFI.

### 2.4. Serum Collection

At 21 and 35 days of age, two goslings per replicate with BWs closest to the replicate average were selected for sampling. For the 21-day-old goslings, 4 mL of blood was collected from the wing vein 12 h after DEX or normal saline injection. The blood samples were allowed to stand at 37 °C for 30 min and then centrifuged at 3000 r/min to separate the serum, which was aliquoted and stored at −80 °C for subsequent analyses.

### 2.5. Determination of Immune Organ Indices

Goslings were euthanized by cervical dislocation, after which the spleen, thymus, and bursa of Fabricius were carefully excised, accurately weighed, and recorded. The immune organ index was calculated using the formula: immune organ index = immune organ weight (g)/live BW (kg). All experimental procedures were conducted following approval from the Animal Ethics Committee of Shanghai Academy of Agricultural Sciences, with measures taken to minimize animal pain during sampling.

### 2.6. HE Staining of Immune Organs

Immune organs, including the bursa of Fabricius, spleen, and thymus, were excised and immediately fixed in 4% paraformaldehyde, with the fixative replaced every 24 h until it became clear to ensure complete tissue fixation. Following fixation, the tissues were dehydrated through a graded ethanol series and then embedded in paraffin. Once the embedding blocks solidified, sections were cut using a microtome and subjected to hematoxylin–eosin staining. The stained sections were subsequently mounted and examined under a microscope to document the histomorphological characteristics of the immune organs.

### 2.7. Morphology of Immune Organs

For the thymus, the cortical/medullary area ratio was selected as the quantitative index. The detection method is as follows: take 4 μm thick HE-stained thymus sections, randomly select 3 three intact thymic lobules per animal under a 200× microscope, use the “polygon selection tool” of ImageJ 1.54g software (Bethesda, MD, USA) to outline and measure the areas of the cortex and medulla, respectively, calculate the ratio of the two, and then take the average value of the three lobules. For the bursa of Fabricius, the number of lymphoid follicles per unit area was chosen as the index. The detection method is as follows: take 4 μm thick HE-stained bursa of Fabricius sections, calibrate the scale (convert pixels to mm^2^) via ImageJ under a 100× microscope, randomly select three non-overlapping fields of view per animal, count the number of intact lymphoid follicles in each field of view, calculate the number of follicles per mm^2^, and take the average value of the three fields of view. For the spleen, the white pulp area percentage was adopted as the index. The detection method is as follows: take 4 μm thick HE-stained spleen sections, select three representative fields of view under a 40× low-power microscope, use ImageJ to mark the white pulp and the total area of the entire field of view, calculate the white pulp area percentage (total white pulp area/total field of view area × 100%), and take the average value of the three fields of view.

### 2.8. Tight Junction Protein Real-Time PCR

Total RNA was isolated from jejunal mucosa using Trizol reagent (Invitrogen, Carlsbad, CA, USA). Prior to reverse transcription, the concentration and purity of the total RNA were quantified with a Nano-drop 1000 Spectrophotometer (Thermo Fisher Scientific, Waltham, MA, USA). Subsequently, the qualified total RNA was reverse-transcribed into complementary DNA (cDNA) using a reverse transcription kit (TransScript, TransGen, Beijing, China).

The β-actin gene was chosen as the internal reference gene for real-time polymerase chain reaction (real-time PCR). This assay was performed to assess the expression profiles of target genes, including zona occludens 1 (ZO-1), Occludin, and Claudin-1. The experiment utilized the TransGen TB Green kit (TransGen Bio Inc., Beijing, China) and a fluorescence quantitative PCR instrument (QuantStudio 5, Thermo Fisher Scientific, Waltham, MA, USA), with specific primer sequences listed in Table 2. For data analysis, the relative quantification method (2^−ΔΔCT^) was employed.

### 2.9. Intestinal Microbiota Sequencing

Cecal chyme samples from the midsection of the intestine were collected and sent to Shenzhen Weishengtai Technology Co., Ltd (Shenzhen, China) for analysis. Total DNA was extracted using the E.Z.N.A. Soil DNA Kit (Omega Bio-Tek, Norcross, GA, USA), and the 16S rRNA V3-V4 region was amplified by PCR using primers 338F (5′-ACTCCTACGGGAGGCAGCAG-3′) and 806R (5′-GGACTACHVGGGTWTCTAAT-3′). Sequence data from different samples were separated using the QIIME2 demux plugin. Following demultiplexing, sequences underwent quality control, trimming, denoising, assembly, and chimera removal using the QIIME2 dada2 plugin to generate the final feature sequences.

### 2.10. Data Analysis

Experimental data were initially organized using Excel and subsequently analyzed by one-way analysis of variance (ANOVA) using SPSS 26.0 (IBM Corp., Armonk, NY, USA). Multiple comparisons were conducted using the least significant difference (LSD) method. All results are presented as mean ± standard error of the mean (SEM), with *p* < 0.05 considered statistically significant.

## 3. Result

### 3.1. Growth Performance

At 14 d, no significant difference in BW was observed among all groups (*p* > 0.05). At 21 d, Con group BW was significantly higher than DEX and DEX + GLP groups (*p* < 0.01). At 35 d, Con group BW remained higher than DEX group (*p* < 0.01) but showed no difference from DEX + GLP group (*p* > 0.05). For ADFI and ADG, at 14–21 d, Con group values were significantly higher than DEX and DEX + GLP groups (*p* < 0.01); at 21–35 d, DEX + GLP group ADFI and ADG were significantly higher than DEX and Con groups (*p* < 0.01). Over 14–35 d, no difference in ADFI and ADG was found between Con and DEX + GLP groups (*p* > 0.05), while both were higher than DEX group (*p* < 0.01). No significant difference in F/G was detected among all groups (*p* > 0.05; Table 3, Figure 1).

### 3.2. Immune Organ Indices

At 21 days of age, the spleen, thymus, and bursa of Fabricius indices in the Con group were significantly higher than those in the DEX and DEX + GLP groups (*p* < 0.01). By 35 days, no significant differences in the indices of the spleen, thymus, or bursa of Fabricius were observed among the groups (*p* > 0.05; Table 4).

### 3.3. Serum Immue Parameters

At 21 days of age, the Con group exhibited significantly higher levels of total protein (TP), albumin (ALB), globulin (GLOB), IL-6, IgA, and IgG compared with the DEX and DEX + GLP groups (*p* < 0.05); there is the same trend for IL-10 and IgM (0.10 > *p* > 0.05). By 35 days, the Con group maintained higher levels of TP, ALB, GLOB, IL-10, and IgG than the DEX group (*p* < 0.05), and IL-6 also showed a similar trend (0.10 > *p* > 0.05), while the DEX + GLP group showed no significant differences from the Con group in TP, ALB, GLOB, and other measured indices. Nevertheless, the Con group still exhibited significantly higher TP, ALB, GLOB, and IgG levels compared with the DEX group (*p* < 0.05; Table 5).

### 3.4. Immune Organ Morphology

Figure 2 shows HE-stained histology of immune organs at 21 and 35 days. At 21 days, Con group had regular plicae, clear crypts, and dense follicles; DEX group showed disordered plicae, atrophied follicles, and widened interstitium; DEX + GLP group had repaired plicae and more follicles. At 35 days, Con group maintained complex plicae; DEX group had sparse plicae and fragmented follicles; DEX + GLP group had restored folds and structure near Con group (Figure 2A). At 21 days, Con group had clear pulp boundaries and intact corpuscles; DEX group had light parenchyma and blurred corpuscles; DEX + GLP group showed darker staining and improved structure. At 35 days, Con group had dense parenchyma; DEX group had fissures; DEX + GLP group recovered density (Figure 2B). At 21 days, Con group had clear lobules and thick cortex; DEX group had atrophied lobules and thin cortex; DEX + GLP group improved lobules. At 35 days, Con group had distinct cortex-medulla; DEX group had sparse parenchyma; DEX + GLP group had thickened cortex and structure near Con group (Figure 2C).

### 3.5. Immune Organ Morphology Indexes

Table 6 analyzes the inter-group differences in thymic cortical/medullary area ratio, lymphoid follicle count per unit area of bursa of Fabricius, and white pulp area percentage of spleen among 21-day-old and 35-day-old goslings in the Con, DEX, and DEX + GLP groups. The results show that on day 21, the thymic cortical/medullary area ratio and lymphoid follicle count per unit area of bursa of Fabricius were significantly higher in the Con group than in the DEX and DEX + GLP groups (*p* < 0.01), with no significant difference in the white pulp area percentage of the spleen; on day 35, the thymic cortical/medullary area ratio was significantly higher in the Con group than in the DEX group (*p* < 0.01), the lymphoid follicle count per unit area of bursa of Fabricius was significantly higher in the Con and DEX + GLP groups than in the DEX group (*p* = 0.023), and there was no significant difference in the white pulp area percentage of the spleen.

### 3.6. Tight Junction Protein mRNA Expression

Figure 3 shows the effect of GLP on DEX-induced expression of jejunal tight junction proteins in goslings (*p* < 0.05): At 21 days of age (Figure 3A), the expression levels of *ZO-1*, *Occludin*, and *Claudin-1* in the Con group were significantly higher than those in the DEX group and DEX + GLP group (*p* < 0.05), while there was no significant difference between the DEX + GLP group and the DEX group. At 35 days of age (Figure 3B), in terms of *ZO-1*, the DEX group was significantly lower than the Con group and DEX + GLP group (*p* < 0.05). In terms of *Occludin*, the Con group was significantly higher than the DEX group (*p* < 0.05), but there was no difference with the DEX + GLP group. There was no significant difference in *Claudin-1* among all groups.

### 3.7. Intestinal Microbiota

At 21 days, α-diversity (Shannon, Pielou’s evenness, Faith’s PD, Simpson) showed no significant differences among groups, but Con group had higher Chao1 and Observed species than DEX and DEX + GLP groups (Figure 4A); β-diversity PCoA (PCo1 = 22.5%, PCo2 = 18.3%) had no group differences (Figure 4B). At 35 days, all α-diversity indices had no group differences (Figure 4C); β-diversity PCoA (PCo1 = 31.7%, PCo2 = 17.7%) showed Con and DEX + GLP groups clustered similarly, while DEX group separated (Figure 4D). At 21 days (phylum level), Proteobacteria (DEX > Con > DEX + GLP) and Firmicutes (Con > DEX + GLP > DEX) dominated; DEX + GLP group had higher Bacteroidetes than Con and DEX groups, and DEX and DEX + GLP groups had higher Cyanobacteria than Con group (Figure 5A). At the genus level, DEX group had highest Cupriavidus and Pseudomonadaceae_Pseudomonas, and DEX + GLP group had higher Flavobacterium than Con and DEX groups (Figure 5B). At 35 days (phylum level), Firmicutes dominated (Con = 75.38%, DEX = 71.95%, DEX + GLP = 77.63%), and DEX group had highest Actinobacteria (Figure 5C). At the genus level, DEX group had highest *Shigella* and *Turicibacter*, and DEX + GLP group had much higher Lactococcus than Con and DEX groups (Figure 5D).

At 21 days (phylum level), Proteobacteria and Firmicutes were core-dominant, followed by Cyanobacteria and Actinobacteria. For Proteobacteria, DEX (49.58%) > Con (41.92%) > DEX + GLP (38.18%); for Firmicutes, Con (27.69%) > DEX + GLP (22.80%) > DEX (21.43%). DEX + GLP group had higher Bacteroidetes (13.75%) than Con (7.18%) and DEX (4.50%); DEX (15.52%) and DEX + GLP (15.05%) groups had higher Cyanobacteria than Con (11.52%) (Figure 5A). At 21 days (genus level), *Cupriavidus* and *Pseudomonadaceae_Pseudomonas* were dominant. For *Cupriavidus*, DEX (15.84%) > Con (11.96%); for *Pseudomonadaceae_Pseudomonas*, DEX (11.94%) > Con (4.25%) and DEX + GLP (7.39%). DEX + GLP group had higher Flavobacterium (9.03%) than Con (1.62%) and DEX (0.37%) (Figure 5B). At 35 days (phylum level), for Firmicutes, Con (75.38%), DEX (71.95%), DEX + GLP (77.63%); for Proteobacteria, Con (21.77%), DEX (23.42%), DEX + GLP (21.13%); for Actinobacteria, DEX (4.52%) > Con (2.51%) > DEX + GLP (0.72%); and for other phyla, <5% (Figure 5C). At 35 days (genus level), for *Shigella*, Con (13.88%), DEX (20.51%), DEX + GLP (16.85%); for *Lactococcus*, DEX + GLP (19.18%) > Con (2.64%) > DEX (0.07%); and for *Turicibacter*, DEX (12.14%) > Con (2.73%) > DEX + GLP (0.43%) (Figure 4D).

### 3.8. Species Composition Differences

LEfSe analysis revealed distinct microbial biomarkers among the groups. At 21 days, the Con group was characterized by f__Peptococcaceae and g__*Peptococcus*, whereas the DEX + GLP group exhibited biomarkers including o__Flavobacteriales, c__Flavobacteria, f__Flavobacteriaceae, g__*Flavobacterium*, p__Bacteroidetes, and g__*Anaerostipes* (Figure 6A). By 35 days, the Con group’s biomarkers comprised f_Weeksellaceae and g__*Clostridium*, the DEX group was marked by f__Enterococcaceae and g__*Enterococcus*, and the DEX + GLP group was characterized by g__*Lactococcus* (Figure 6B).

## 4. Discussion

As a natural functional additive, plant polysaccharides hold broad application potential in animal husbandry and are expected to serve as ideal alternatives to synthetic chemical additives by enhancing production performance. Studies have shown that dietary supplementation with Chinese herbal polysaccharides, including Yupingfeng polysaccharide, ginseng polysaccharide, and artichoke polysaccharide, is widely used to improve growth performance, immunity, and meat quality in poultry [16,17,18]. The efficacy of polysaccharides in promoting poultry growth is primarily attributed to their multiple bioactive mechanisms. First, polysaccharides can regulate the appetite control system by modulating the expression of appetite-related genes in the hypothalamus, stimulating appetite-promoting factors such as neuropeptide Y, and suppressing satiety signals like leptin, thereby increasing feed intake and providing a sufficient nutritional basis for growth [19]. Second, polysaccharides enhance intestinal health and digestive function: they act as prebiotics to stimulate beneficial gut bacteria, inhibit pathogenic colonization, and maintain microbiota balance, while also promoting intestinal villus development, increasing villus height and absorptive area, strengthening the mucosal barrier, and reducing nutrient loss [20,21]. Together, these effects mitigate growth inhibition caused by stress and disease, ultimately improving growth performance by increasing feed intake, enhancing nutrient utilization efficiency, and reducing growth loss. Among various natural polysaccharides, GLP, as representative active compounds, have demonstrated notable efficacy in enhancing poultry growth performance [22,23].

This study demonstrated that GLP exerted a stage-specific regulatory effect on DEX-induced growth inhibition in goslings. At 21 days, the BW of the Con group was significantly higher than that of the DEX and DEX + GLP groups, indicating the pronounced inhibitory effect of DEX and the absence of a detectable GLP intervention effect, likely due to the strong initial impact of DEX or the need for cumulative action of GLP. By 35 days, although the Con group remained heavier than the DEX group, no significant difference was observed between the Con and DEX + GLP groups, suggesting that with extended intervention, GLP gradually mitigated the inhibitory effects of DEX, restoring BW in the DEX + GLP group toward normal levels and reflecting its positive regulatory potential. Between 21 and 35 days, the DEX + GLP group exhibited significantly higher ADFI and ADG than both the DEX and Con groups, indicating that in the later stage, GLP not only alleviated growth inhibition but also enhanced growth beyond the normal group, likely by promoting feeding behavior and nutrient utilization, potentially through modulation of appetite or metabolism [19]. No significant differences in F/G were observed among groups, consistent with previous studies reporting that GLP does not generally improve growth via feed conversion efficiency [24,25]. Nonetheless, in the DEX-induced stress model, GLP clearly alleviated growth suppression, particularly during the later phase, by enhancing feed intake and nutrient utilization, counteracting stress-induced growth inhibition, and even surpassing normal growth. This effect may be linked to the targeted regulation of GLP on stress-related metabolic disturbances, as stress redirects energy toward defensive metabolism, whereas GLP may reallocate energy to growth pathways by alleviating oxidative stress and related mechanisms [19,26]. In conclusion, the growth-promoting effect of GLP is not broad-spectrum but provides specific alleviation in the context of stress-induced growth inhibition.

In the early post-hatch stage, poultry exhibit weak environmental adaptability and low disease resistance, making the enhancement of immunity critical for preventing infectious diseases [27]. Immune organs serve as specialized tissues that are essential for the production, maturation, and regulation of immune cells during immune responses [28]. The development and maturation of these organs are vital for effective immune function, and lymphocyte development is commonly used as an indicator of immune status. Generally, larger immune organ weights and higher organ indices are associated with stronger humoral and cellular immune capabilities [29].

Studies have demonstrated that Astragalus polysaccharides exert immunomodulatory effects on both central and peripheral immune organs, including the bone marrow, thymus, lymph nodes, spleen, and mucosal tissues [30,31]. Similarly, Hericium erinaceus polysaccharides have been shown to mitigate immune organ damage induced by Muscovy duck reovirus infection while enhancing antioxidant capacity [32].

This study investigated the protective effects of GLP on DEX-induced immune organ damage in goslings. Comparative analysis revealed that GLP partially reversed the organ development inhibition and tissue damage caused by DEX, with this protective effect being particularly pronounced at 21 days. These findings not only confirmed the immunomodulatory properties of GLP but also suggested potential multi-dimensional mechanisms, including enhancement of antioxidant capacity, activation of immune cells, and regulation of intestinal microbiota [33]. Regarding immune organ indices, DEX-induced organ atrophy observed in this study reflected the inhibitory effects of stress on lymphocyte development, potentially mediated by hormonal interference with signaling pathways [34]. GLP intervention significantly increased these indices, possibly by activating the NF-κB pathway to inhibit apoptosis, thereby promoting lymphocyte proliferation and maturation [35]. Supporting this view, recent studies have shown that supplementation with G. lucidum powder can enhance immune performance in mice [36]. Interestingly, differences among groups narrowed by 35 days, indicating the age-dependent self-repair capacity of the poultry immune system; however, the early intervention with GLP accelerated immune recovery, highlighting its advantage in promoting early immune protection.

The structural integrity of immune organs underpins their proper function, and morphological changes directly reflect immune status [37]. In this study, tissue morphological observations further elucidated the immune injury caused by DEX and the regulatory role of GLP. At 21 days, DEX induced pronounced microstructural damage, including disordered and broken plicae with atrophied and reduced follicles in the bursa of Fabricius, decreased red pulp cell density and blurred splenic corpuscles in the spleen, and thinned cortex with disorganized interlobular septa in the thymus. These alterations represent more than morphological changes; they directly impair the microenvironment essential for immune cell production and maturation. The bursal follicles are critical for B lymphocyte differentiation and maturation [38], the thymic cortex is central for T cell development, and splenic red pulp and corpuscles are involved in immune cell migration and immune response initiation [39]. Consequently, DEX-induced structural damage hindered lymphocyte development, correlating with observed decreases in immune organ indices and serum immune factors, confirming that early immune suppression by DEX is mediated through microstructural disruption [40]. GLP intervention demonstrated significant immunomodulatory effects by repairing organ morphology. At 21 days, the DEX + GLP group already exhibited morphological improvements, such as increased bursal follicle numbers, deeper splenic parenchymal staining, and higher thymic cell density, indicating early protection of immune structure. By 35 days, these effects were more pronounced, with restored plica folds and aggregated follicles in the bursa of Fabricius, reduced fissures and increased parenchymal density in the spleen, and thickened cortex with lobular structure approaching normal status in the thymus. These structural restorations provided a supportive microenvironment for immune cell proliferation and differentiation—intact follicles facilitated B cell maturation, a thickened thymic cortex promoted T cell development, and dense splenic parenchyma ensured efficient initiation of immune responses [41]. This linkage between structural repair and functional recovery also explains why immune organ indices and serum immune factor levels in the DEX + GLP group approached those of the control by 35 days, highlighting the potential of GLP as an effective immunomodulator.

The intestinal microbiota, as a crucial component of the poultry immune system, plays a key role in maintaining host health, modulating immune responses, and mitigating stress-induced damage [11]. Evidence indicates that microbiota diversity and composition directly influence immune organ development and disease resistance, for instance, by producing short-chain fatty acids (SCFAs) that regulate inflammatory pathways or enhance intestinal barrier function [42]. In DEX-induced stress models, dysbiosis of the intestinal microbiota often contributes to immunosuppression [43], whereas natural polysaccharides like GLP can provide protection by reshaping microbial communities [44,45]. In this study, *Lactococcus* emerged as a biomarker of the DEX + GLP group at 35 days, highlighting its potential role in alleviating DEX-induced intestinal and immune injury in goslings. As a common intestinal probiotic, its high abundance can exert protective effects through multiple mechanisms. For maintaining microecological balance, *Lactococcus* secretes metabolites such as lactic acid and bacteriocins, lowering intestinal pH, inhibiting the colonization and proliferation of opportunistic pathogens such as *Shigella* (enriched in the DEX group), and reducing inflammation risk [46], consistent with the observed microbiota structure of the DEX + GLP group closely resembling that of the control at 35 days (minimal β-diversity differences). Additionally, *Lactococcus* enhances nutrient metabolism efficiency by effectively decomposing carbohydrates and proteins, thereby improving feed digestion and absorption and providing greater energy and amino acids for the host [47]. This may underpin the recovery of serum TP and globulin levels in the DEX + GLP group, forming a “intestinal–nutrition–immune” linkage that supports immune function. Furthermore, *Lactococcus* can interact with intestinal mucosal cells to promote the proliferation of mucosal immune cells, such as lymphocytes, and stimulate anti-inflammatory factor secretion, indirectly alleviating immunosuppression [48]. These effects align with the observed improvements in immune organ morphology and serum antibody levels, suggesting that *Lactococcus* may act as a key intermediary in the “intestinal microbiota–immune regulation” axis mediating GLP’s immunomodulatory effects.

## 5. Conclusions

GLP effectively mitigated DEX-induced immune injury in goslings by enhancing growth performance, restoring the structure and function of immune organs, regulating serum immune factors, and reshaping the intestinal microbiota, particularly through the enrichment of *Lactococcus*.

## Figures and Tables

**Figure 1 animals-15-03226-f001:**
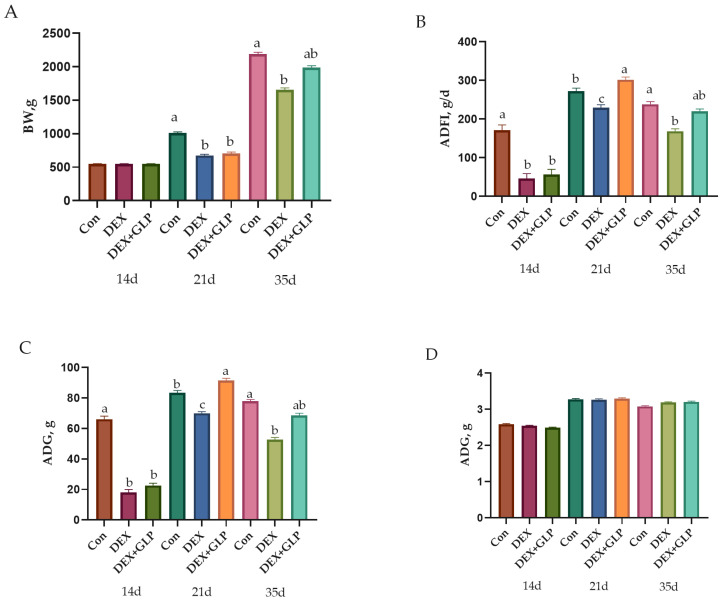
Growth performance metrics showing (**A**) body weight, (**B**) average daily feed intake (**C**) average daily gain, (**D**) feed conversion ratio. ^a, b, c^ Different letter superscripts mean significant differences (*p* < 0.05).

**Figure 2 animals-15-03226-f002:**
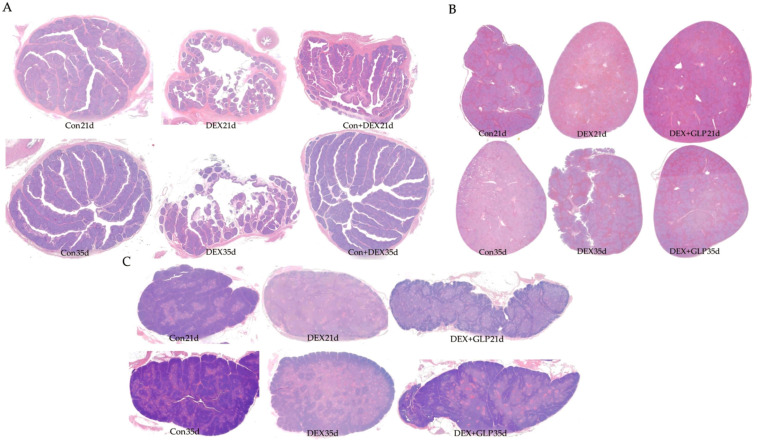
Effects of GLP on the immune organs of goslings induced by DEX: (**A**) bursa of Fabricius, (**B**) thymus, (**C**) spleen.

**Figure 3 animals-15-03226-f003:**
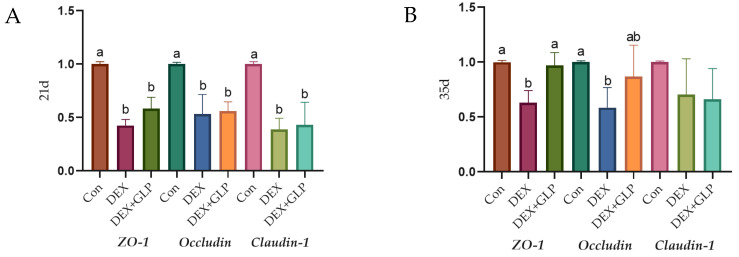
The effect of GLP on DEX-induced tight junction protein in goslings. (**A**): 21d, (**B**): 35d. ^a, b^ Different letter superscripts mean significant differences (*p* < 0.05).

**Figure 4 animals-15-03226-f004:**
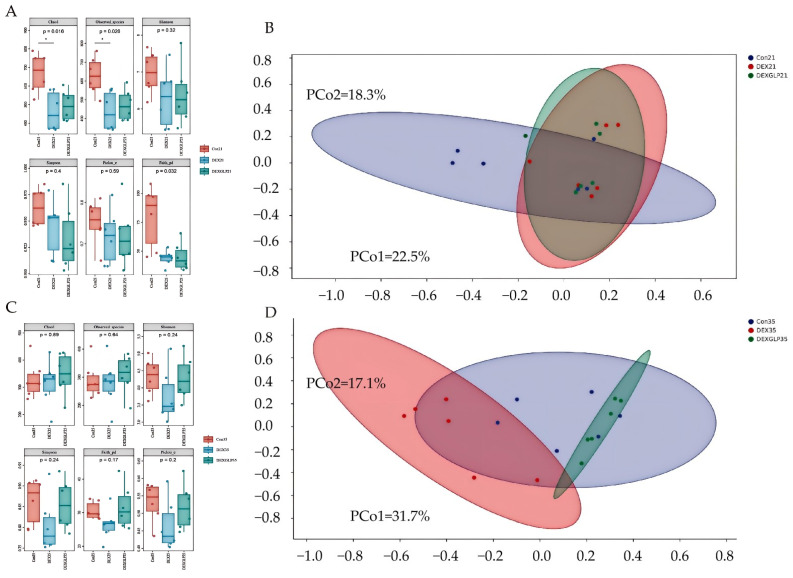
The effect of GLP on DEX-induced cecal microbiota in goslings. (**A**): 21d α diversity, (**B**): 21d, β diversity, (**C**): 35d α diversity, (**D**): 35d β diversity. Note: * Indicates statistical significance at the 0.05 level.

**Figure 5 animals-15-03226-f005:**
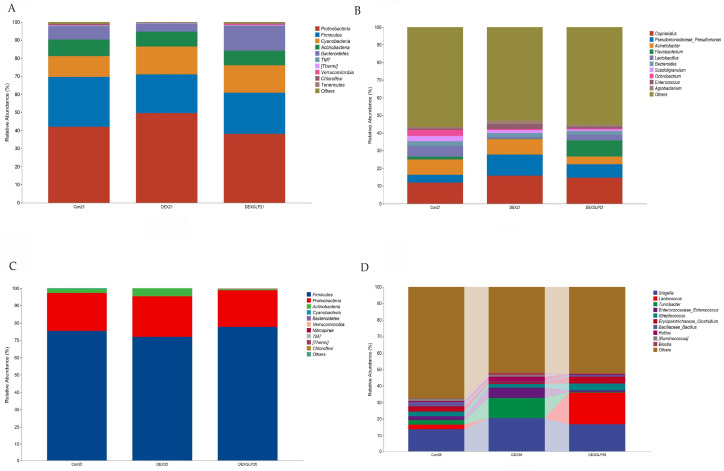
The effect of GLP on DEX-induced cecal microbiota in goslings. (**A**): 21d phylum-level composition, (**B**): 21d belongs to genus-level composition, (**C**): 35d phylum-level composition, (**D**): 35d genus-level composition.

**Figure 6 animals-15-03226-f006:**
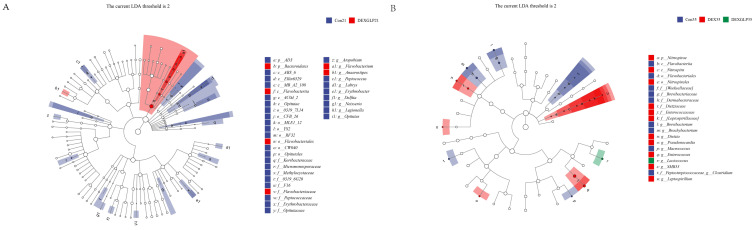
Linear discriminant analysis (LDA) (LDA > 2.0, *p* < 0.05) of cecal microbiomes. (**A**): 21d. (**B**): 35d. Note: Different color shades indicate the relative abundance size of microbial taxa; Deeper colors mean a higher LDA score for the taxon in its group.

**Table 1 animals-15-03226-t001:** Composition and nutrient level of experiment diets (air-dry basis).

Items	Content (%)
Ingredients	
Corn	67.92
Soybean meal (43% crude protein)	24.90
Soybean oil	2.00
Lys (98%)	0.09
Met (98%)	0.09
Vitamin-mineral Premix ^1^	5.00
Total	100.00
Nutrient level ^2^	
CP	16.00
ME (MJ/kg)b	12.40
CF	2.56
Ca	0.79
P	0.51
Lys	0.90
Met	0.45
Thr	0.63
Cys	0.21

^1^ One kilogram of the premix contained the following: NaCl 4 g, Fe 100 mg, Cu 8 mg, Mn 120 mg, Zn 100 mg, Se 0.4 mg, Co 1.0 mg, I 0.4 mg, VA 8330 IU, VB1 2.0 mg, VB2 0.8 mg, VB6 1.2 mg, VB12 0.03 mg, VD3 1440 IU, VE 30 IU, biotin 0.2 mg, folic acid 2.0 mg, calcium pantothenic acid 20 mg, niacin acid 40 mg. ^2^ CP (Crude protein), ME (Metabolizable energy), CF (Crude fiber).

**Table 2 animals-15-03226-t002:** Primer sequences for genes used in qPCR.

Genes	Primer Sequence 5′-3′	Genbank
ZO-1	F:CTAGCTAGCGTACAGTACAC	*XM_013177404.1*
	R:CTCTCTCATAGGCAGGAAAC	
Occludin	F:GCTGGGCTACAACTACGGGT	*XM_013199669*
	R:ACGATGGAGGCGATGAGC	
Claudin-1	F:GGAAGATGACCAGGTGAAG	*XM_013199194.1*
	R:GGAAGATGACCAGGTGAAG	
β-actin	F:TCCGTGACATCAAGGAGAAG	*XM_013174886.1*
	R:CATGATGGAGTTGAAGGTGG	

**Table 3 animals-15-03226-t003:** The effect of GLP on the growth performance of goslings induced by DEX.

Items	Groups				
	Con	DEX	DEX + GLP	SEM	*p*-Value
BW, g					
14d	547.89	549.50	548.72	3.99	0.969
21d	1011.75 ^a^	675.25 ^b^	708.00 ^b^	14.37	<0.01
35d	2187.50 ^a^	1657.19 ^b^	1987.58 ^ab^	22.14	<0.01
ADFI, g/d					
14–21d	171.31 ^a^	45.99 ^b^	56.71 ^b^	10.93	<0.01
21–35d	272.55 ^b^	229.54 ^c^	301.94 ^a^	6.90	<0.01
14–35d	238.80 ^a^	168.36 ^b^	220.20 ^ab^	5.98	<0.01
ADG, g					
14–21d	66.27 ^a^	17.96 ^b^	22.75 ^b^	1.95	<0.01
21–35d	83.39 ^b^	70.14 ^c^	91.40 ^a^	1.15	<0.01
14–35d	78.08 ^a^	52.75 ^b^	68.52 ^ab^	1.01	<0.01
F/G					
14–21d	2.59	2.54	2.49	0.03	0.706
21–35d	3.28	3.27	3.30	0.04	0.965
14–35d	3.08	3.19	3.21	0.04	0.418

Note: BW (body weight), ADG (average daily gain), ADFI (average daily feed intake), F/G (feed/gain ratio). Con = control group; DEX = dexamethasone group; DEX + GLP = dexamethasone + Ganoderma lucidum polysaccharides group. ^a, b, c^ Different letter superscripts mean significant differences (*p* < 0.05).

**Table 4 animals-15-03226-t004:** The effect of GLP on the immune organ of goslings induced by DEX.

Items	Groups				
	Con	DEX	DEX + GLP	SEM	*p*-Value
21d					
Spleen index, g/kg	1.38 ^a^	0.88 ^b^	1.04 ^b^	0.05	<0.01
Thymus index, g/kg	1.80 ^a^	0.67 ^b^	0.71 ^b^	0.12	<0.01
Bursa of Fabricius index, g/kg	1.02 ^a^	0.77 ^b^	0.61 ^b^	0.05	0.007
35d					
Spleen index, g/kg	1.37	0.95	1.16	0.06	0.077
Thymus index, g/kg	1.44	1.18	1.36	1.35	0.529
Bursa of Fabricius index, g/kg	0.74	0.50	0.60	0.05	0.384

Note: Con = control group; DEX = dexamethasone group; DEX + GLP = dexamethasone + Ganoderma lucidum polysaccharides group. ^a, b^ Different letter superscripts mean significant differences (*p* < 0.05).

**Table 5 animals-15-03226-t005:** The effect of GLP on the serum immune of goslings induced by DEX.

Items	Groups				
	Con	DEX	DEX + GLP	SEM	*p*-Value
21d					
TP, g/kg	115.02 ^a^	99.77 ^bc^	95.10 ^c^	1.80	<0.01
ALB, g/kg	58.13 ^a^	49.73 ^b^	48.60 ^b^	0.34	0.033
GLOB, g/kg	57.92 ^a^	50.68 ^b^	46.88 ^b^	1.23	0.001
IL-6, pg/mL	37.42 ^a^	28.79 ^b^	28.00 ^b^	1.23	0.004
IL-10, pg/mL	48.89	37.67	48.53	1.84	0.059
IgA, μg/mL	369.39 ^a^	330.07 ^ab^	293.48 ^b^	11.55	0.013
IgG, g/L	24.67 ^a^	18.59 ^b^	19.03 ^b^	0.78	0.004
IgM, μg/mL	1993.68	1501.72	1809.22	74.43	0.078
35d					
TP, g/kg	115.81 ^a^	92.55 ^b^	108.55 ^a^	2.10	<0.01
ALB, g/kg	58.58 ^a^	47.88 ^b^	51.76 ^ab^	1.26	0.005
GLOB, g/kg	57.05 ^a^	44.97 ^b^	57.62 ^a^	1.48	0.002
IL-6, pg/mL	40.95	33.35	38.04	0.90	0.078
IL-10, pg/mL	50.68 ^a^	39.64 ^b^	50.33 ^a^	1.55	0.017
IgA, μg/mL	363.13	305.64	327.58	9.80	0.227
IgG, g/L	23.02 ^a^	17.82 ^b^	21.19 ^ab^	0.69	0.047
IgM, μg/mL	1697.64	1546.24	1612.66	70.62	0.769

Note: Con = control group; DEX = dexamethasone group; DEX + GLP = dexamethasone + Ganoderma lucidum polysaccharides group. ^a, b, c^ Different letter superscripts mean significant differences (*p* < 0.05).

**Table 6 animals-15-03226-t006:** The effect of GLP on the *immune organ morphology* of goslings induced by DEX.

Items	Groups				
	Con	DEX	DEX + GLP	SEM	*p*-Value
21d					
Thymic Cortical/Medullary Area Ratio	1.84 ^a^	0.99 ^b^	1.23 ^b^	0.07	<0.01
Lymphoid Follicle Count per Unit Area of Bursa of Fabricius (units/mm^2^)	18.02 ^a^	10.23 ^b^	12.74 ^b^	0.61	<0.01
White Pulp Area Percentage of Spleen (%)	20.12	17.23	18.23	0.59	0.745
35d					
Thymic Cortical/Medullary Area Ratio	1.89 ^a^	1.25 ^b^	1.74 ^ab^	0.09	<0.01
Lymphoid Follicle Count per Unit Area of Bursa of Fabricius (units/mm^2^)	22.69 ^a^	16.78 ^b^	23.14 ^a^	0.76	0.023
White Pulp Area Percentage of Spleen (%)	23.52	19.74	21.35	1.12	0.145

Note: Con = control group; DEX = dexamethasone group; DEX + GLP = dexamethasone + Ganoderma lucidum polysaccharides group. ^a, b^ Different letter superscripts mean significant differences (*p* < 0.05).

## Data Availability

The sequencing data generated and analyzed during the current study are available in the National Center for Biotechnology Information (NCBI) Sequence Read Archive (SRA) under the accession number PRJNA1348303 (http://www.ncbi.nlm.nih.gov/bioproject/1348303, accessed on 24 October 2025). Other data associated with this study are presented within the manuscript. For any additional data requirements, please contact the corresponding author, Daqian He, with a reasonable request.

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
