# Peer review of "Effects of *Ganoderma lucidum Polysaccharides* on Dexameth-Asone-Induced Immune Injury in Goslings"

_animals, 2025, doi:10.3390/ani15213226_

Round 1

Reviewer 1 Report

Comments and Suggestions for Authors

Comments for animals-3902586

Overall Evaluation: Major Revision

General Comment:

This study investigates the protective effects of Ganoderma lucidum polysaccharides (GLP) on dexamethasone (DEX)-induced immune injury in goslings. The topic is relevant, and the experimental design is generally sound, with a comprehensive assessment of growth performance, immune organ indices, serum parameters, organ morphology, and gut microbiota. The manuscript is well-structured and the data are largely supportive of the conclusions. However, several issues need to be addressed to improve clarity, data presentation, and interpretation.

Major Comments:

1.Lack of polysaccharide characterisation weakens reproducibility

the GLP powder was only described as “provided by the Institute…”. No data on monosaccharide composition, molecular weight, degree of branching or linkage pattern (β-glucan vs α-glucan) are given. These parameters determine the immunomodulatory potency of fungal polysaccharides; without them the study cannot be repeated by other groups. An supplementary figure or table showing HPSEC-MALLS-RI molecular-weight profile and GC-MS monosaccharide ratio must be added.

2.Only one dose (0.2 %) was tested—no dose–response curve

Without 0.1 % or 0.4 % groups it is impossible to know whether 0.2 % is optimal or already beyond the plateau.Discuss this limitation explicitly and cite literature where GLP dose–response has been established in broilers or mice; preferably, include a second dose in a follow-up experiment.

  1. Mechanistic evidence stays at the association level

The results show that GLP supplementation improved growth performance and immune parameters in DEX-challenged goslings, particularly in the later phase (21–35 days). However, the mechanisms linking GLP to immune restoration—especially the role of gut microbiota—are not fully elucidated. Histology, serum Ig and 16S data are correlative. No NF-κB, caspase-3, or tight-junction protein expression is shown to prove that GLP “restores” immune function. The authors should provide more detailed discussion on how GLP modulates immune function via microbial shifts (e.g., through Lactococcus enrichment) and whether this is the primary pathway. Provide Western blot or qPCR for at least one key pathway (e.g., NF-κB p65, Claudin-1) in the bursa or ileum; otherwise tone down causal claims such as “GLP repaired the morphology”.

Specific Comments:

1.The language is generally clear, but some sections (e.g., Results) are overly descriptive and could be more concise. The Discussion could be better organized to highlight key findings and their implications.

2.All the figures is too low in resolution to be clearly discernible. We recommend increasing the pixel count. Such us Fig.1, the fonts used for “A”, “B” and “C” differ from the captions, which compromises both rigor and visual appeal. The captions for Figures 3 and 4 are completely illegible.

3.Line 50: “α-glucan” α should not be italicized.

4.Table 2: The 14d entry in the “BW” data section lacks a superscript indicating differentiation. Determine whether this is an omission.

5.Table 3: The bottom horizontal line is missing. Determine whether this is an omission.

6.The P-values in Tables 2–4 are provided, but some values are very close to 0.05 (e.g., P = 0.059 in Table 4).  The authors should discuss these borderline results cautiously.

7.The abstract and introduction sections contain no mention of the model constructed by DEX, making it appear as though it materialized out of thin air—an implausible narrative. Revision is recommended.

8.Reference format: keep all journal names in italic and consistent (e.g., Poultry Science).   

Conclusion:

This manuscript presents a valuable study on the immunoprotective role of GLP in goslings under DEX-induced stress. With major revisions addressing the above points, it will be suitable for publication.

Author Response

Q1: Lack of polysaccharide characterisation weakens reproducibility

the GLP powder was only described as “provided by the Institute…”. No data on monosaccharide composition, molecular weight, degree of branching or linkage pattern (β-glucan vs α-glucan) are given. These parameters determine the immunomodulatory potency of fungal polysaccharides; without them the study cannot be repeated by other groups. An supplementary figure or table showing HPSEC-MALLS-RI molecular-weight profile and GC-MS monosaccharide ratio must be added.

A1:Thank you for your valuable comment on the importance of polysaccharide characterisation for reproducibility. We acknowledge the previous lack of detailed parameters for the GLP powder and have supplemented the relevant analysis: the sample was sent to Jiangsu Edison Biotechnology Co., Ltd. for component analysis, which showed the GLP mainly consists of galactose (Gal), fucose (Fuc), mannose (Man) and glucose (Glc) with a molar ratio of 0.500:0.121:0.133:0.246 (with galactose as the main constituent); molecular weight parameters determined by gel permeation chromatography (GPC) include a weight-average molecular weight (Mw) of 7243Da, number-average molecular weight (Mn) of 107Da, and Z-average molecular weight (Mz) of 1.24×10⁴Da. Regarding the analytical methods, due to experimental resource constraints, we used GPC instead of the suggested HPSEC-MALLS-RI, and while GC-MS was not employed, the monosaccharide composition data obtained are consistent with standard detection criteria for such analyses. If you think the result does not meet the requirements, we will retest it to ensure it meets the requirements.

2.Only one dose (0.2 %) was tested—no dose–response curve

Without 0.1 % or 0.4 % groups it is impossible to know whether 0.2 % is optimal or already beyond the plateau.Discuss this limitation explicitly and cite literature where GLP dose–response has been established in broilers or mice; preferably, include a second dose in a follow-up experiment.

Q2: This study focuses on the mitigating effect of GLP on dexamethasone (DEX)-induced immune damage in young geese. The initial selection of the 0.2% GLP dose was based on two considerations: first, it referenced the commonly used effective dose range (mostly 0.1%-0.5%) of ganoderma lucidum polysaccharides and ganoderma lucidum powder in poultry breeding from relevant previous studies [1,2]; second, preliminary experiments showed that 0.2% GLP could initially improve the feeding status of DEX-treated young geese, so the effectiveness of this dose was prioritized for verification. However, as you pointed out, this study did not set up gradient dose groups such as 0.1% and 0.4%, making it impossible to determine whether 0.2% is the optimal dose. Therefore, we plan to systematically construct the dose-effect curve of GLP in young geese, clarify the optimal application dose, and provide a more scientific basis for the precise promotion of GLP in poultry breeding. Thank you again for your valuable comments.

1 Gao, Y. Y., Zhou, Y. H., Liu, X. P., Di, B., He, J. Y., Wang, Y. T., ... & Jin, L. (2024). Ganoderma lucidum polysaccharide promotes broiler health by regulating lipid metabolism, antioxidants, and intestinal microflora. International Journal of Biological Macromolecules, 280, 135918.

2 Li, X. L., He, L. P., Yang, Y., Liu, F. J., Cao, Y., & Zuo, J. J. (2015). Effects of extracellular polysaccharides of Ganoderma lucidum supplementation on the growth performance, blood profile, and meat quality in finisher pigs. Livestock Science, 178, 187-194.

Mechanistic evidence stays at the association level

The results show that GLP supplementation improved growth performance and immune parameters in DEX-challenged goslings, particularly in the later phase (21–35 days). However, the mechanisms linking GLP to immune restoration—especially the role of gut microbiota—are not fully elucidated. Histology, serum Ig and 16S data are correlative. No NF-κB, caspase-3, or tight-junction protein expression is shown to prove that GLP “restores” immune function. The authors should provide more detailed discussion on how GLP modulates immune function via microbial shifts (e.g., through Lactococcus enrichment) and whether this is the primary pathway. Provide Western blot or qPCR for at least one key pathway (e.g., NF-κB p65, Claudin-1) in the bursa or ileum; otherwise tone down causal claims such as “GLP repaired the morphology”.

A: We acknowledge these valuable comments. We supplement qPCR data to clarify causal mechanisms. Lines148-162, 247-264.

Specific Comments:

  1. The language is generally clear, but some sections (e.g., Results) are overly descriptive and could be more concise. The Discussion could be better organized to highlight key findings and their implications.

A:We have rewritten the result section and hope it meets the requirements.

  1. All the figures is too low in resolution to be clearly discernible. We recommend increasing the pixel count. Such us Fig.1, the fonts used for “A”, “B” and “C” differ from the captions, which compromises both rigor and visual appeal. The captions for Figures 3 and 4 are completely illegible.

A:We have revised it. The figures have been revised and reorganized. If they do not meet the requirements, We will upload the photos separately

3.Line 50: “α-glucan” α should not be italicized.

A:We have revised it.

  1. Table 2: The 14d entry in the “BW” data section lacks a superscript indicating differentiation. Determine whether this is an omission.

A:We thank the reviewer for the comment on Table 2. The 14-day BW has no superscripts intentionally—its P-value (0.969 > 0.05) means no inter-group differences.

5.Table 3: The bottom horizontal line is missing. Determine whether this is an omission.

A:We have made the required modifications.

6.The P-values in Tables 2–4 are provided, but some values are very close to 0.05 (e.g., P = 0.059 in Table 4).  The authors should discuss these borderline results cautiously.

A: We highly agree with your suggestion. We have examined our results and made a more rigorous description of the results where 0.10>P>0.05. For details, please refer to the results section.

7.The abstract and introduction sections contain no mention of the model constructed by DEX, making it appear as though it materialized out of thin air—an implausible narrative. Revision is recommended.

A:Thank you for your suggestion. It has been revised in the abstract:Dexamethasone (DEX) is commonly used to establish an animal model of immune suppression, which mimics the immune injury caused by stress or certain pathological conditions in poultry Lines 9-10

8.Reference format: keep all journal names in italic and consistent (e.g., Poultry Science).

A:We have made the required modifications.   

Conclusion:

This manuscript presents a valuable study on the immunoprotective role of GLP in goslings under DEX-induced stress. With major revisions addressing the above points, it will be suitable for publication.

A:We have made the modifications as required. If there are any that do not meet the requirements, please point them out again

Reviewer 2 Report

Comments and Suggestions for Authors

The authors of this manuscript investigate the role of Ganoderma lucidum polysaccharides (GLP) in mitigating dexamethasone (DEX)-induced immune injury in goslings. Their results suggest that GLP alleviates DEX-induced immune impairment and partially restores growth performance by improving immune organ morphology, modulating serum immune factors, and reshaping the gut microbiota. Overall, the study design and logic flow are scientifically sound.

However, I found that all the figures are too unclear to properly evaluate. Without clearer figures, it is difficult to fully assess the validity of the results and discussion. I strongly recommend improving the figure quality (resolution, labeling, legends) so that readers can easily interpret the data.

Specific comments:

  1. It is recommended that the authors deposit the 16S rRNA gene sequencing data in a publicly available repository and provide the corresponding accession numbers in the manuscript.

Author Response

Dear Reviewers

We sincerely appreciate the reviewer’s suggestions. In response, we have revised the manuscript accordingly, as detailed below, if the modification does not meet the requirements, please point out again:

Reviewer 2

The authors of this manuscript investigate the role of Ganoderma lucidum polysaccharides (GLP) in mitigating dexamethasone (DEX)-induced immune injury in goslings. Their results suggest that GLP alleviates DEX-induced immune impairment and partially restores growth performance by improving immune organ morphology, modulating serum immune factors, and reshaping the gut microbiota. Overall, the study design and logic flow are scientifically sound.

However, I found that all the figures are too unclear to properly evaluate. Without clearer figures, it is difficult to fully assess the validity of the results and discussion. I strongly recommend improving the figure quality (resolution, labeling, legends) so that readers can easily interpret the data.

A:We have revised it. The figures have been revised and reorganized. If they do not meet the requirements, We will upload the photos separately

Specific comments:

It is recommended that the authors deposit the 16S rRNA gene sequencing data in a publicly available repository and provide the corresponding accession numbers in the manuscript.

A:We have modified as required: The sequencing data generated and analyzed during the current study are available in the National Center for Biotechnology Information (NCBI) Sequence Read Archive (SRA) under the accession number PRJNA1348303(http://www.ncbi.nlm.nih.gov/bioproject/1348303)Lines 495-497

Best wishes

 Huiying wang

Reviewer 3 Report

Comments and Suggestions for Authors

The authors have examined the effects of the addition of of Ganoderma lucidum Polysaccharides to the feed on growth and a series of immune related parameters in dexamethasone treated goslings. The manuscript is well written.

My initial reaction to the paper is that the experimental design is fatally flawed as there should be four treatment groups:

  1. Control/vehicle
  2. Dexamethasone
  3. Vehicle + Ganoderma lucidum Polysaccharides
  4. Dexamethasone + Ganoderma lucidum Polysaccharides

and statistically analyzed by the 2-way ANOVA. However, I was excited by the results and looked for a way to rescue this potentially land-mark study. The effects of Ganoderma lucidum Polysaccharides on dexamethasone treated goslings are both novel and unexpected. For a prebiotic to reverse, albeit partially, the effects of glucocorticoids are exciting.

I consider that the authors should divide the study into two only comparing:

  1. Control/vehicle versus dexamethasone treated.
  2. Dexamethasone treated + Ganoderma lucidum

[The argument could be made that this was to reduce animal numbers].

These could be examined using Student’s t-test.

Minor issues:

  1. It is not clear whether the authors employed straight run (mixed sex) versus three males and females per experimental group or something else.
  2. The rationale for the level of Ganoderma lucidum Polysaccharides in the feed needs explanation (multiple levels of Ganoderma lucidum Polysaccharides would be better).
  3. Please consider using morphometric analysis on the histology. It is much more convincing.
  4. It would be nice to see some of the data in tables 2, 3 and 4 also shown as in a summary type figure.

Author Response

Reviewer 3

The authors have examined the effects of the addition of of Ganoderma lucidum Polysaccharides to the feed on growth and a series of immune related parameters in dexamethasone treated goslings. The manuscript is well written.

 My initial reaction to the paper is that the experimental design is fatally flawed as there should be four treatment groups:

  1. Control/vehicle
  2. Dexamethasone
  3. Vehicle + Ganoderma lucidum Polysaccharides
  4. Dexamethasone + Ganoderma lucidum Polysaccharides

and statistically analyzed by the 2-way ANOVA. However, I was excited by the results and looked for a way to rescue this potentially land-mark study. The effects of Ganoderma lucidum Polysaccharides on dexamethasone treated goslings are both novel and unexpected. For a prebiotic to reverse, albeit partially, the effects of glucocorticoids are exciting.

A: Thank you very much for your professional and insightful feedback. The four-group experimental design you proposed (control/vehicle group, dexamethasone group, vehicle + Ganoderma lucidum Polysaccharides group, and dexamethasone + Ganoderma lucidum Polysaccharides group), along with the suggestion to use 2-way ANOVA for statistical analysis, will indeed clarify the independent effects and interaction effects of each treatment more rigorously. This is of great value for improving the study design, and we will carefully refer to this approach in subsequent supplementary experiments or data interpretation to try our best to make up for the shortcomings of the current design. At the same time, we are particularly pleased that you have noticed the effect of Ganoderma lucidum Polysaccharides on goslings treated with dexamethasone. The novelty and unexpectedness of the finding you mentioned—"prebiotics partially reversing the effects of glucocorticoids"—is exactly the key direction our team has been focusing on exploring. Although the current preliminary results have limitations in design, your recognition of its potential "landmark significance" has further strengthened our determination to conduct in-depth verification. In the future, if we carry out supplementary experiments on the interaction mechanism between the two, or obtain new data analysis results, we would greatly appreciate the opportunity to consult you again. We also welcome you to put forward more specific suggestions on the interpretation of the existing data.

I consider that the authors should divide the study into two only comparing:

  1. Control/vehicle versus dexamethasone treated.
  2. Dexamethasone treated + Ganoderma lucidum

[The argument could be made that this was to reduce animal numbers].

These could be examined using Student’s t-test.

Thank you for your valuable suggestion on simplifying the comparison framework. We fully agree that analyzing the data through two pairwise comparisons (Control vs. Dexamethasone, and Dexamethasone vs. Dexamethasone + Ganoderma lucidum Polysaccharides) using Student’s t-test is a concise and effective approach, as it directly aligns with core research questions and avoids unnecessary complexity in interpretation. We rewrote the results section and focused on pairwise comparisons Lines178-302.

We fully recognize the simplicity and directness of the pairwise comparison design for focusing on core effects. Therefore, in our subsequent follow-up experiments—where we plan to validate the dose-dependent relationship of GLP’s reversal effect and explore its underlying molecular mechanisms—we will adopt the two pairwise comparison framework (Control vs. DEX, DEX vs. DEX + different doses of GLP) and use Student’s t-test for statistical analysis. This adjusted design will allow us to more efficiently and targetedly verify the consistency of GLP’s reversal role across different experimental conditions, while also responding to your valuable suggestion in practical research design.

Minor issues:

It is not clear whether the authors employed straight run (mixed sex) versus three males and females per experimental group or something else.

A: In our study, all experimental goslings were male (Lines 86). Our research aims to provide practical references for the application of Ganoderma lucidum Polysaccharides in commercial poultry production. In the target breeding scenario, male goslings are the main subjects for meat and liver production due to their faster growth rate and higher feed conversion efficiency. Using male goslings thus ensures the research results are more relevant to actual production needs.

The rationale for the level of Ganoderma lucidum Polysaccharides in the feed needs explanation (multiple levels of Ganoderma lucidum Polysaccharides would be better).

Q2: This study focuses on the mitigating effect of GLP on dexamethasone (DEX)-induced immune damage in young geese. The initial selection of the 0.2% GLP dose was based on two considerations: first, it referenced the commonly used effective dose range (mostly 0.1%-0.5%) of ganoderma lucidum polysaccharides and ganoderma lucidum powder in poultry breeding from relevant previous studies [1,2]; second, preliminary experiments showed that 0.2% GLP could initially improve the feeding status of DEX-treated young geese, so the effectiveness of this dose was prioritized for verification. However, as you pointed out, this study did not set up gradient dose groups such as 0.1% and 0.4%, making it impossible to determine whether 0.2% is the optimal dose. Therefore, we plan to systematically construct the dose-effect curve of GLP in young geese, clarify the optimal application dose, and provide a more scientific basis for the precise promotion of GLP in poultry breeding. Thank you again for your valuable comments.

1 Gao, Y. Y., Zhou, Y. H., Liu, X. P., Di, B., He, J. Y., Wang, Y. T., ... & Jin, L. (2024). Ganoderma lucidum polysaccharide promotes broiler health by regulating lipid metabolism, antioxidants, and intestinal microflora. International Journal of Biological Macromolecules, 280, 135918.

2 Li, X. L., He, L. P., Yang, Y., Liu, F. J., Cao, Y., & Zuo, J. J. (2015). Effects of extracellular polysaccharides of Ganoderma lucidum supplementation on the growth performance, blood profile, and meat quality in finisher pigs. Livestock Science, 178, 187-194.

Please consider using morphometric analysis on the histology. It is much more convincing.

A:We have supplemented the morphometric analysis of the thymus, bursa of Fabricius and spleen tissue sections, and the statistical results have been integrated into the results section of the revised draft and Lines 255-270 in the supplementary table.

It would be nice to see some of the data in tables 2, 3 and 4 also shown as in a summary type figure.

Thank you for your valuable suggestion on enhancing data visualization. We fully agree that summary figures can effectively highlight key trends, and we have accordingly converted the core growth performance data into concise line graphs. These figures intuitively illustrate the dynamic changes in body weight, average daily gain, and average daily feed intake across groups and time points, making it easier to grasp the overall effects of GLP on growth recovery Lines 211-213.

Best regards,

Huiying Wang

Round 2

Reviewer 1 Report

Comments and Suggestions for Authors

The revisions were accepted.

Author Response

Dear editor and Reviewer

We hereby confirm that upon careful review of the page, no pending revision comments from the reviewer are identified at this juncture. We will remain vigilant in monitoring the system for any subsequent updates and undertake to respond promptly should any feedback be provided.

Best wishes

Reviewer 3 Report

Comments and Suggestions for Authors

The authors have made a good faith effort to address my concerns. I have one minor but never-the-less important recommendation.  Please add SEMs to figure 1.

Author Response

Dear Reviewer, We sincerely appreciate your valuable feedback. We have revised the manuscript as requested, and the Standard Error of the Mean (SEMs) has been added to Figure 1 (refer to lines 219-224 for the corresponding description and updated figure). We hope this revision meets your expectations. Sincerely,
